# A cluster randomized trial assessing the effect of a digital health algorithm on quality of care in Tanzania (DYNAMIC study)

Rainer Tan[1,2,3,4☯*], Godfrey Kavishe[5☯†], Alexandra V. Kulinkina[3,4], Sabine Renggli[2], Lameck B. Luwanda[2], Chacha Mangu[5], Geofrey Ashery[2], Margaret Jorram[2], Ibrahim Evans Mtebene[2], Peter Agrea[5], Humphrey Mhagama[5], Kristina Keitel[3,4,6], Marie-Annick Le Pogam[1], Nyanda Ntinginya[5‡], Honorati Masanja[2‡], Valérie D'Acremont[1,3,4‡]

1 Centre for Primary Care and Public Health (Unisanté), University of Lausanne, Lausanne, Switzerland, 2 Ifakara Health Institute, Dar es Salaam, United Republic of Tanzania, 3 Swiss Tropical and Public Health Institute, Allschwil, Switzerland, 4 University of Basel, Basel, Switzerland, 5 National Institute of Medical Research–Mbeya Medical Research Centre, Mbeya, United Republic of Tanzania, 6 Pediatric Emergency Department, Department of Pediatrics, University Hospital Bern, Bern, Switzerland

☯ These authors contributed equally to this work.
† Deceased.
‡ NN, HM and VD also contributed equally to this work.
* rainer.tan@unisante.ch

**Data Availability Statement:** De-identified data can be found at https://zenodo.org/records/10849644.

## Abstract

Digital clinical decision support tools have contributed to improved quality of care at primary care level health facilities. However, data from real-world randomized trials are lacking. We conducted a cluster randomized, open-label trial in Tanzania evaluating the use of a digital clinical decision support algorithm (CDSA), enhanced by point-of-care tests, training and mentorship, compared with usual care, among sick children 2 to 59 months old presenting to primary care facilities for an acute illness in Tanzania (ClinicalTrials.gov NCT05144763). The primary outcome was the mean proportion of 14 major Integrated Management of Childhood Illness (IMCI) symptoms and signs assessed by clinicians. Secondary outcomes included antibiotic prescription, counseling provided, and the appropriateness of antimalarial and antibiotic prescriptions. A total of 450 consultations were observed in 9 intervention and 9 control health facilities. The mean proportion of major symptoms and signs assessed in intervention health facilities was 46.4% (range 7.7% to 91.7%) compared to 26.3% (range 0% to 66.7%) in control health facilities, an adjusted difference of 15.1% (95% confidence interval [CI] 4.8% to 25.4%). Only weight, height, and pallor were assessed statistically more often when using the digital CDSA compared to controls. Observed antibiotic prescription was 37.3% in intervention facilities, and 76.4% in control facilities (adjusted risk ratio 0.5; 95% CI 0.4 to 0.7; p<0.001). Appropriate antibiotic prescription was 81.9% in intervention facilities and 51.4% in control facilities (adjusted risk ratio 1.5; 95% CI 1.2 to 1.8; p = 0.003). The implementation of a digital CDSA improved the mean proportion of IMCI symptoms and signs assessed in consultations with sick children, however most symptoms and signs were assessed infrequently. Nonetheless, antibiotics were prescribed less often, and more appropriately. Innovative approaches to overcome barriers related to clinicians' motivation and work environment are needed.

**Funding:** This study was supported by a grant from the Fondation Botnar Switzerland (https://www.fondationbotnar.org/) grant number 6278 to V.D.A., and from the Swiss Development Cooperation (https://www.fdfa.admin.ch/sdc) project number 7F-10361.01.01 to V.D.A.. The study sponsor (Centre for Primary Care and Public Health, Unisanté, University of Lausanne) led the study design, the writing of the report and the decision to submit the article for publication. The funders of the study had no role in study design, data collection, data analysis, data interpretation or writing of the report.

**Competing interests:** The authors have declared that no competing interests exist.

## Author summary

Digital health tools have been created to help healthcare workers provide better care, but their real-world impact is still uncertain. This study evaluated the use of ePOCT+, a digital health clinical decision support algorithm for health providers, combined with point-of-care diagnostic tools (CRP tests, pulse oximetry), training, and mentorship. The study compared 9 primary care health facilities using ePOCT+ with 9 facilities providing care as usual. The study found that using the digital health intervention improved the assessment of important symptoms and signs in children aged 2 to 59 months with acute illnesses. However, many symptoms and signs were still not frequently assessed. In addition, antibiotic prescriptions were halved in facilities using the tool, and the appropriateness of prescriptions improved significantly. Despite these benefits, challenges related to healthcare workers' motivation and work environments must be explored to fully realize the potential of such tools. These findings suggest that digital health tools can improve the quality of care and address issues like overprescription of antibiotics. However, broader strategies are needed to support healthcare workers in delivering comprehensive, high-quality care in similar settings globally.

## Introduction

Millions of preventable deaths are attributed to suboptimal healthcare quality [1]. Factors such as staff shortages, inadequate budget allocation, poor clinical knowledge, and limited access to quality medical education, mentorship and supervision collectively contribute to this issue [2–4]. In response to this challenge, and to reduce childhood mortality, the World Health Organization developed the Integrated Management of Childhood Illness (IMCI) Chartbook [5]. Since its inception, over 100 countries have implemented the guidelines, and IMCI may reduce mortality and improve quality of care [6,7]. However, poor adherence to IMCI is common, limiting its benefits [8–10].

Digital Clinical Decision Support Algorithms (CDSAs) were devised to enhance adherence to clinical guidelines. These tools, typically operating on electronic tablets or mobile phones, guide healthcare providers through the consultation process, by prompting the evaluation of symptoms, signs, and recommended diagnostic tests, to finally propose the appropriate diagnosis and treatment [11,12]. While several studies have found that using these digital CDSAs improve adherence to IMCI, a noteworthy research gap is that many of these investigations were conducted in controlled study settings, and most lacked randomization [13–20].

ePOCT+, a digital CDSA, was developed based on insights from two previous generations of CDSAs [21,22], specifically addressing challenges by our CDSAs and others, such as limited scope and information technology difficulties [23]. The present study aimed to assess whether this CDSA associated with point-of-care tests, training, and mentorship, would improve the quality of care for sick children compared to usual care, by comparing adherence to IMCI in a pragmatic cluster randomized trial.

## Methods

### Study design

The present study is an open-label, parallel-group, cross-sectional cluster randomized trial within the DYNAMIC Tanzania project. An external clinical researcher observed a sample of consultations from health facilities from both study arms documenting adherence of

healthcare providers to quality-of-care indicators. The study was a planned ancillary study within a larger cluster randomized trial conducted between 1 December 2021 and 31 October 2022 using a sample of the clusters [24]. A cluster design was chosen since the intervention was targeted at the health facility and healthcare provider. The trial design and rationale are outlined in the protocol available in the parent trial registration on ClinicalTrials.gov number NCT05144763 and in the supplementary materials (S1 File). The detailed statistical analysis plan for this ancillary study is also available in the supplementary materials (S2 File).

The study design and implementation were collaboratively executed between both Tanzanian (Ifakara Health Institute, National Institute for Medical Research—Mbeya Medical Research Centre) and Swiss (Centre for Primary Care and Public Health [Unisanté]–University of Lausanne, and Swiss Tropical and Public Health Institute) partners. The design was guided by input from patients, and health providers during the implementation of similar trials in Tanzania [14,22,25]. Over 100 community engagement meetings involving over 7,000 participants were conducted before and during the study. These meetings included discussions with Community and Regional Health Management Teams in Tanzania.

## Participants

The health facility was the unit of randomization since the intervention targeted both the healthcare provider and health facility. Primary care health facilities (dispensaries or health centers) were eligible for inclusion if they performed on average 20 or more consultations with children 2 months to 5 years per week, were government or government-designated health facilities, and were located less than 150 km from the research institutions. Specific to this study, consultations were only included if healthcare providers had been trained to use ePOCT +, and the ePOCT+ tool was functioning on the day of observation (no IT issues related to power outages, or crashes reported).

In contrast to the larger trial that included children aged 1 day to 14 years, this ancillary study included only children aged 2 to 59 months old presenting for an acute medical or surgical condition at participating health facilities. Children presenting solely for scheduled consultations for a chronic disease (e.g. HIV, tuberculosis, malnutrition), for routine preventive care (e.g. growth monitoring, vaccination), or a follow-up consultation were excluded.

The study was conducted in 5 councils within the Mbeya and Morogoro regions of Tanzania, with two councils being semi-urban and three rural. Malaria prevalence in febrile children was low in three councils, and moderately high in two. HIV prevalence among children less than 5 years old in Tanzania is 0.4% [26]. Healthcare for children under 5 years of age is free for all acute illnesses at government or government-designated primary health facilities, including the cost of medications [27]. Nurses and clinical officers routinely provide outpatient care in dispensaries, while in health centers medical doctors sometimes provide care as well. Clinical Officers, the predominant healthcare providers at primary health facilities, are non-physician health professionals with 2–3 years of clinical training following secondary school [28].

## Interventions

The intervention involved equipping health facilities with ePOCT+, an electronic clinical decision support algorithm on an Android based tablet (Fig 1), along with associated point-of-care tests (C-Reactive Protein, Hemoglobin, pulse oximetry), training, and mentorship. ePOCT+ prompts the healthcare provider to answer questions about demographics, symptoms, signs, and tests [23]. Based on the answers, ePOCT+ proposes one or more diagnoses, treatments, and management plans including referral recommendation. Healthcare providers had the

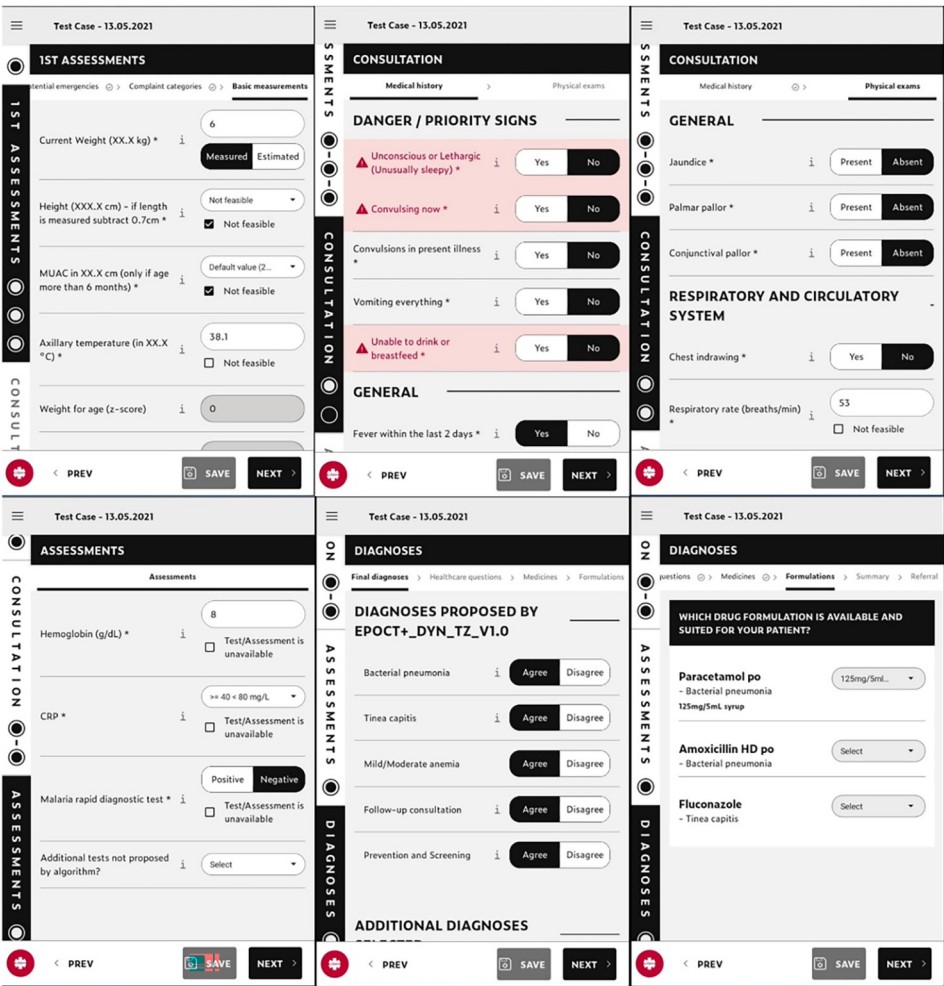

**Fig 1. Screen-shots of different stages of ePOCT+ running on the medAL-reader application.** Stages are shown in order of appearance, however not all stages are shown in the figure.

possibility to deviate from ePOCT+ recommendations, and had the option to not use the tool. In order to move forward within the different sections of the digital tool, it was mandatory to respond to all IMCI symptoms and signs, except for height and mid-upper arm circumference (MUAC) which was optional. The tool allowed some signs to be estimated (temperature, respiratory rate) or based on recent measurements (weight). Detailed description on the development process and features of ePOCT+ and the medAL-reader application can be found in separate publications [23,29].

The implementation team provided mentorship to intervention health facilities. This mentoring consisted of regular visits to health facilities every 2–3 months, and frequent communication via phone calls or group messages (3–4 times per month) to address issues, offer guidance, and gather feedback on the new tools. Quality-of-care dashboards were shared through group messages, enabling healthcare providers to compare their antibiotic prescription rates, uptake, and other quality-of-care indicators with other facilities (benchmarking). Control health facilities continued with usual care, did not have access to clinical data dashboards, and only received visits from the implementation team to help resolve issues related to the electronic case report forms (eCRFs).

The infrastructure provided to all health facilities (control and intervention) included a tablet for each outpatient consultation room, a router, a local server (Raspberry Pi), internet connectivity, and backup power (battery or solar system if needed). If unavailable weighing scales, mid-upper arm circumference (MUAC) bands, and thermometers were provided to all health facilities. Healthcare providers from both intervention and control facilities underwent equivalent clinical refresher training on IMCI and concepts of antibiotic stewardship. Additionally, specific training was provided on the use of the ePOCT+ CDSA in intervention facilities and the use of the eCRF in control facilities.

## Outcomes

The primary outcome was the mean proportion of 14 pre-identified major IMCI symptoms and signs assessed by the healthcare provider, as observed by an external clinical research assistant. The included symptoms were fever, cough or difficult breathing, convulsions in this illness, diarrhea, ear pain or discharge, child unable to drink or breastfeed, and child vomits everything. The included signs were measurement of temperature, respiratory rate, pallor, weight, mid-upper arm circumference (MUAC), height, and skin turgor. In specific circumstances, some patients were not included in the denominator for specific signs as they were not clinically indicated as defined by IMCI: they include MUAC in children less than 6 months old, respiratory rate in the absence of cough or difficult breathing, and skin turgor in the absence of diarrhea. If cough or difficult breathing was not assessed, then we took the most conservative approach assuming that respiratory rate should have been measured, and the same for diarrhea and skin turgor. Of note "lethargic and unconscious" was considered as assessed if the clinician asked the caregiver if it was present during the illness, and not based on observation of the child as being "lethargic or unconscious".

Secondary outcomes include the proportion of consultations during which each major IMCI symptom and sign were assessed, the proportion of which other symptoms and signs were assessed (S2 File), the proportion of consultations where different IMCI counseling was conducted, and proportion of consultations for which antibiotics were prescribed. The rationale for the distinction between "major" IMCI and "other" symptoms and signs are described in detail in the statistical analysis plan (S2 File). Prescription of antibiotics was assessed by the research assistant by observing the actual prescription prescribed. The appropriateness of antibiotic prescription in relation to the retained diagnosis was also assessed. An antibiotic prescription was considered appropriate if one of the retained diagnoses required an antibiotic as per IMCI or the WHO hospital pocket book, and the absence of a prescription if no diagnosis required an antibiotic [30,31]. An appropriate antimalarial prescription was a prescription of any antimalarial if there was a positive malaria test. Assessment of appropriateness was conducted blinded to the study arm.

All outcomes pertained to the cluster level (health facility), and were assessed by an external clinical research assistant who observed the consultations in the consultation room without interfering with the consultation. The external clinical research assistants were clinical officers with experience in primary care consultations for children. Data was collected using a structured and pre-tested observation form programmed on ODK, and collected on an Android-based tablet. The observation form was based on the 2012 DHS Service Provision Assessment Survey form [32]. Of note, a more recent modification to this survey was developed after the initial planning of this study [32]. Modifications were made to the survey form, to shorten the duration of the evaluation and align it more closely with the aim of the evaluation, to incorporate additional signs and symptoms in line with IMCI 2014 guidelines such as duration of symptoms and the symptomatic assessment of lethargic or unconscious, and additions used by similar evaluations conducted previously [17].

Initially, antibiotic prescription was considered a co-primary outcome alongside the current primary outcome (proportion of 14 major IMCI symptoms and signs) but was later reclassified as a secondary outcome. We made this change to focus the analysis on quality of care, given that antibiotic prescription was already the primary outcome of the large longitudinal cluster randomized trial [24].

## Sample size

The original sample size calculation was based on the previous co-primary outcome of antibiotic prescription. To detect a 25% absolute decrease in mean antibiotic prescription from a baseline of 50%, using an intraclass correlation coefficient (ICC) of 0.10 and an alpha of 0.05, a sample size of 25 patients in 9 clusters (health facilities) per arm was required to have 80% power. The ICC was based on studies evaluating prescription variations among different health care facilities/practices, ranging from 0.07 to 0.10 [33–36].

Expecting a high variability in baseline values of symptoms and signs assessed by a clinician and between clinicians [8,13,17,37], the above sample size would have 67–93% power to detect a 30% absolute increase in the assessment of major IMCI symptoms and signs, considering a baseline value of 40–60%, an ICC of 0.15–0.25, and an alpha of 0.05.

## Randomization

Within the parent trial, health facilities were randomized 1:1 by an independent statistician, stratified by monthly attendance, type of health facility (dispensary or health center), region, and council [24]. For the present ancillary study, another independent statistician sampled 18/ 40 facilities to be included. This included 8/8 health centers (4 intervention, and 4 control), and 10/32 dispensaries. Among the 32 dispensaries, 10 were randomly sampled, stratified by study arm and region (following the same 3:2 ratio in favor of the Morogoro region as done in the parent trial). Due to the nature of the intervention, it was not feasible to blind the healthcare providers, patients, study implementers, or external clinical research assistants (observers) to the intervention.

A convenience sampling was employed, whereby the external clinical researcher observed all eligible consultations while present at the health facility during normal standard working hours (Monday to Friday, 8:00 to 15:00).

## Statistical methods

All analyses were performed using an intention-to-treat approach, i.e. all children with a recorded outcome were included in the analysis regardless if the intervention, ePOCT+, was used or not. The eCRF was designed to prevent missing data, as such all data was complete. All analyses were performed using a clustered-level analysis approach instead of an individual-level analysis due to the small number of clusters included [38,39]. Such an approach has been shown to be robust in cluster randomized trials with less than 30 clusters, with cluster sizes of >10, when assessing outcomes with a prevalence of > 10%, and performs best when cluster sizes are similar, and in case of high ICC [38,39]. The analysis was performed using a two-stage approach as outlined by Hayes et al [39,40], to adjust for both the cluster-level and individual-level covariates. In the first stage, we used a logistic regression model for binary outcomes and a linear regression model for continuous outcomes adjusting for covariates and ignoring clustering and trial arm. Cluster-level residuals were then calculated for each cluster. In the second stage, the residuals were compared to estimate risk ratios for the binary outcomes and mean risk difference for the continuous outcome (including the primary outcome) between study arms. Pre-specified cluster-level covariates were the type of health facility,

council, healthcare worker cadre and healthcare worker years of experience. Pre-specified individual-level covariates of patients were age and sex. Covariates were pre-specified based on their potential influence on the study outcomes. All analyses were performed using Stata v16, v17 and v18 [41].

## Ethics

Written informed consent was obtained from all parents or guardians of participants when attending the participating health facility during the enrollment period. We also requested, written informed consent from all healthcare providers for which their consultations were observed during this ancillary study. Ethical approvals were granted from the Ifakara Health Institute (IHI/IRB/No: 11–2020), the Mbeya Medical Research Ethics Committee *(SZEC-2439/ R.A/V.1/65)*, the National Institute for Medical Research Ethics Committee (NIMR/HQ/R.8a/ Vol. IX/3486 and NIMR/HQ/R.8a/Vol. IX/3583) in Tanzania and from the cantonal ethics review board of Vaud (CER-VD 2020–02800) in Switzerland.

## Results

### Baseline characteristics of health facilities, healthcare providers, and patients

Between 23 March 2022 to 3 June 2022, 225 consultations were observed in 9 intervention facilities, and 225 consultations in 9 control facilities (Fig 2). The type of health facility, urban/ rural localization, and region were well distributed between both study arms (Table 1). A total of 17 healthcare providers saw patients in the control arm, and 22 in the intervention arm during the study. Distribution of sex, age, working experience and cadre of healthcare providers were similar in both study arms. The number of healthcare providers with less than 3 years of experience was slightly higher in the control arm. Among the included patients, there were slightly more female patients and median age was slightly higher in the intervention arm

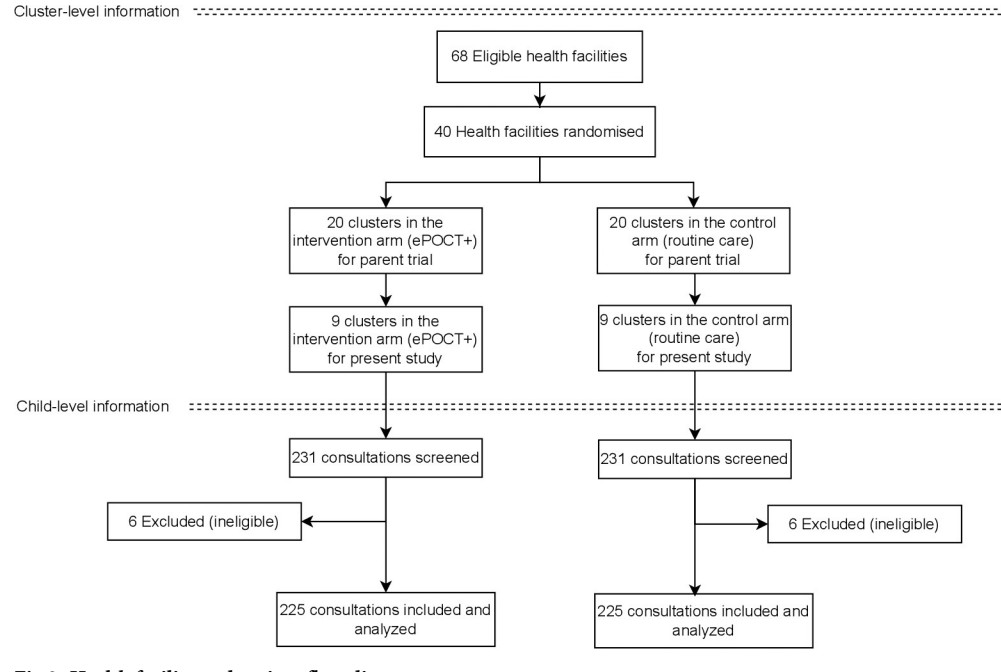

**Fig 2. Health facility and patient flow diagram.**

**Table 1. Characteristics of health facilities, healthcare providers, and patients.**

| Characteristics of health facilities | | Control (N = 9) | Intervention (N = 9) |
|---|---|---|---|
| Type of facility | Dispensary n (%) | 5 (56%) | 5 (56%) |
| | Health Center n (%) | 4 (44%) | 4 (44%) |
| Geographical distribution | Urban n (%) | 4 (44%) | 3 (33%) |
| | Rural n (%) | 5 (56%) | 6 (67%) |
| Region | Mbeya n (%) | 4 (44%) | 4 (44%) |
| | Morogoro n (%) | 5 (56%) | 5 (56%) |
| **Characteristics of healthcare providers** | | **Control (N = 17)** | **Intervention(N = 22)** |
| Sex | Female n (%) | 9 (53%) | 9 (41%) |
| | Male n (%) | 8 (47%) | 13 (59%) |
| Age | Years (Median; IQR) | 32 (28,36) | 34 (30,38) |
| | 20- <30 years n (%) | 5 (29%) | 4 (18%) |
| | 30- <40 years n (%) | 9 (53%) | 14 (64%) |
| | 40- <50 years n (%) | 2 (12%) | 3 (14%) |
| | 50- <60 years n (%) | 1 (6%) | 1 (5%) |
| Experience* | < = 3 years | 7 (41%) | 6 (27%) |
| | 3–5 years | 3 (18%) | 6 (27%) |
| | 5–10 years | 3 (18%) | 6 (27%) |
| | >10 years | 4 (24%) | 4 (18%) |
| Cadre | Medical Doctor | 3 (18%) | 4 (18%) |
| | Assistant Medical Officer | 1 (6%) | 1 (5%) |
| | Clinical Officer | 9 (53%) | 10 (46%) |
| | Clinical Assistant | 0 (0%) | 1 (5%) |
| | Registered or Enrolled Nurse | 4 (24%) | 4 (18%) |
| | Medical Attendant | 0 (0%) | 2 (9%) |
| **Characteristics of patients** | | **Control (N = 225)** | **Intervention (N = 225)** |
| Sex | Female n (%) | 108 (48%) | 123 (55%) |
| | Male n (%) | 117 (52%) | 102 (45%) |
| Age | Months (Median; IQR) | 14 (7,28) | 19 (9,31) |
| | 2–11 months n (%) | 98 (44%) | 69 (31%) |
| | 12–23 months n (%) | 54 (24%) | 72 (32%) |
| | 24–35 months n (%) | 33 (15%) | 36 (16%) |
| | 36–47 months n (%) | 24 (11%) | 26 (12%) |
| | 48–59 months n (%) | 16 (7%) | 22 (10%) |

IQR: Interquartile range

* Experience: Years of working experience

compared to the control. Within the intervention arm, ePOCT+ was used throughout the whole consultation in 213/225 (95%) of consultations, partially used in 5/225 (2%), used after the consultation in 6/225 (3%), and not used at all in 1/225 (0.4%) of consultations.

## Assessment of symptoms and signs, and counseling

The primary outcome of mean proportion of major IMCI symptoms and signs assessed was higher by an adjusted difference of 15.1% (95% confidence interval [CI] 4.8% to 25.4%), p-value 0.007) in intervention health facilities (mean of 46.4%, range 7.7% to 91.7%) compared to control health facilities (mean of 26.3%, range 0% to 66.7%) (Table 2 and Fig 3). Weight, mid-upper arm circumference (MUAC) and pallor were the only individual assessments

**Table 2. Major IMCI Symptoms and Signs Assessed.**

| Primary Outcome | Control, mean % (range) | Intervention, mean % (range) | Intraclass correlation coefficient | Adjusted mean difference with 95% CI[a] | p-value |
|---|---|---|---|---|---|
| *Primary outcome* | | | | | |
| Major IMCI Symptoms and Signs | 26.3% (0%; 66.7%) | 46.4% (7.7%; 91.7%) | 0.652 | 15.1% (4.8%; 25.4%) | **0.007** |
| Secondary Outcome | **Control, n/N[a] (%)** | **Intervention, n/N[b] (%))** | Intraclass correlation coefficient | **Adjusted risk ratio with 95% CI[a]** | **p-value** |
| IMCI Symptoms assessed: | | | | | |
| Convulsions in this illness[b] | 16/225 (7.1%) | 75/225 (33.3%) | 0.442 | 2.1 (0.6; 7.0) | 0.208 |
| Unable to drink or breastfeed[b] | 47/225 (20.9%) | 107/225 (47.6%) | 0.275 | 2.3 (0.8; 6.4) | 0.109 |
| Vomiting everything[b] | 41/225 (18.2%) | 81/225 (36.0%) | 0.524 | 1.7 (0.5; 5.9) | 0.363 |
| Fever | 196/225 (87.1%) | 204/225 (90.7%) | 0.060 | 1.0 (1.0; 1.14) | 0.342 |
| Cough or difficulty breathing | 188/225 (83.6%) | 189/225 (84.0%) | 0.098 | 1.0 (0.9; 1.2) | 0.847 |
| Diarrhea | 111/225 (49.3%) | 130/225 (57.8%) | 0.289 | 1.1 (0.5; 2.4) | 0.872 |
| Ear problem | 11/225 (4.9%) | 37/225 (16.4%) | 0.504 | 1.0 (0.3; 3.5) | 0.951 |
| IMCI Signs assessed | | | | | |
| Weight | 38/225 (16.9%) | 128/225 (56.9%) | 0.582 | 4.9 (1.9; 12.9) | **0.004** |
| Height | 1/225 (0.4%) | 3/225 (1.3%) | 0.059 | 0.3 (0.1; 2.1) | 0.225 |
| MUAC | 4/195 (2.1%) | 131/202 (64.9%) | 0.719 | 5.5 (1.7; 17.6) | **0.008** |
| Temperature | 95/225 (42.2%) | 148/225 (65.8%) | 0.586 | 1.9 (0.6; 5.6) | 0.227 |
| Pallor | 13/225 (5.8%) | 72/225 (32.0%) | 0.324 | 4.1 (1.6; 10.4) | **0.005** |
| Respiratory rate | 15/182 (8.2%) | 47/164 (28.7%) | 0.280 | 1.9 (0.6; 6.1) | 0.230 |
| Skin turgor | 4/153 (2.6%) | 16/141 (11.4%) | 0.142 | 2.1 (0.9; 5.0) | 0.087 |

CI: Confidence interval; MUAC: Mid-upper arm circumference

[a]The differences and relative risk were adjusted by type of health facility, council, healthcare worker cadre, healthcare worker years of experience, patient age and sex

[b]Denominator based on patients for which this sign is clinically indicated to measure, i.e. for MUAC only children age 6 months and above, for respiratory rate only patients for which cough or difficulty is present or not asked, etc

[c]IMCI Danger sign

among the pre-identified major IMCI symptoms and signs that showed a statistically significant difference in the adjusted risk ratio (Table 2 and Fig 4). Among other symptoms and signs assessed, there was a statistically significant difference in the proportion of patients for which the duration of cough, duration of diarrhea, mother's HIV status were assessed, and the proportion of children who were undressed for the physical examination (Table 3). There was no statistical difference in the proportion of counseling topics covered by healthcare providers between study arms (Table 4). For most outcomes the intraclass correlation coefficient (ICC) was relatively high suggesting high variability in adherence to IMCI between health facilities. For the primary outcome of major IMCI symptoms and signs assessed, the ICC was higher in intervention health facilities (0.608) compared to control health facilities (0.354). This difference can also be seen when visualizing the mean proportion of major IMCI symptoms and signs assessed from each health facility in a scatterplot (Fig 3).

## Antibiotic and antimalarial prescription

Antibiotic prescription as observed by the external clinical researcher was lower in intervention health facilities compared to control health facilities with an adjusted risk ratio of 0.5, 95% CI 0.4 to 0.7 p-value <0.001 (Table 5 and Fig 5). The documented antibiotic prescription by healthcare providers (in the ePOCT+ tool for intervention facilities, and eCRF in control

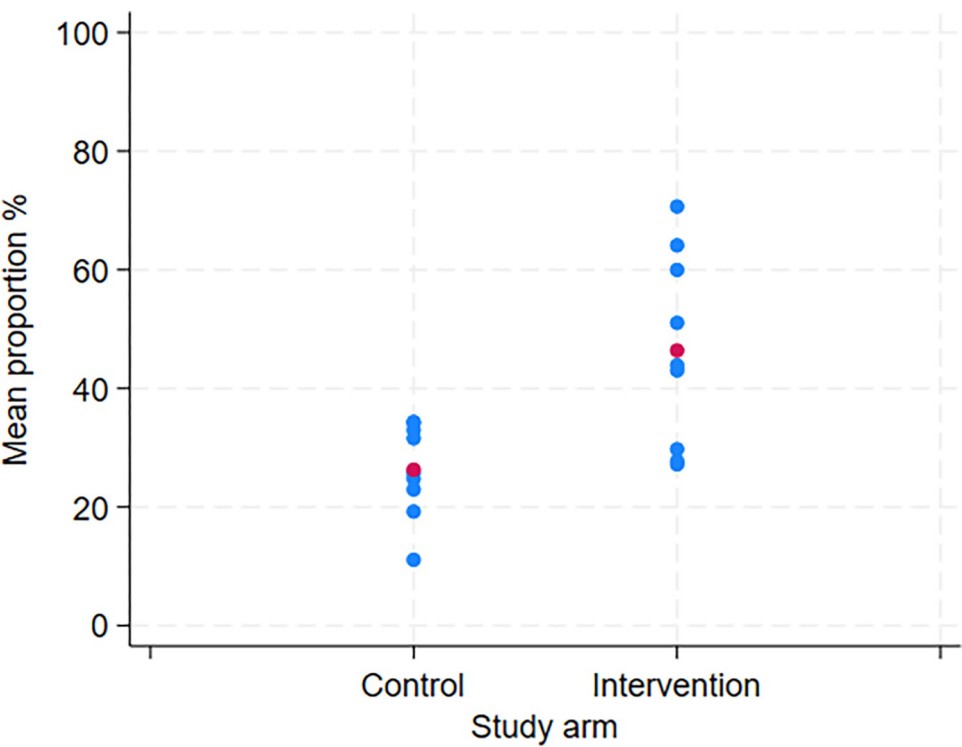

**Fig 3. Scatter-plot of the mean proportion of the major IMCI symptoms and signs assessed.**

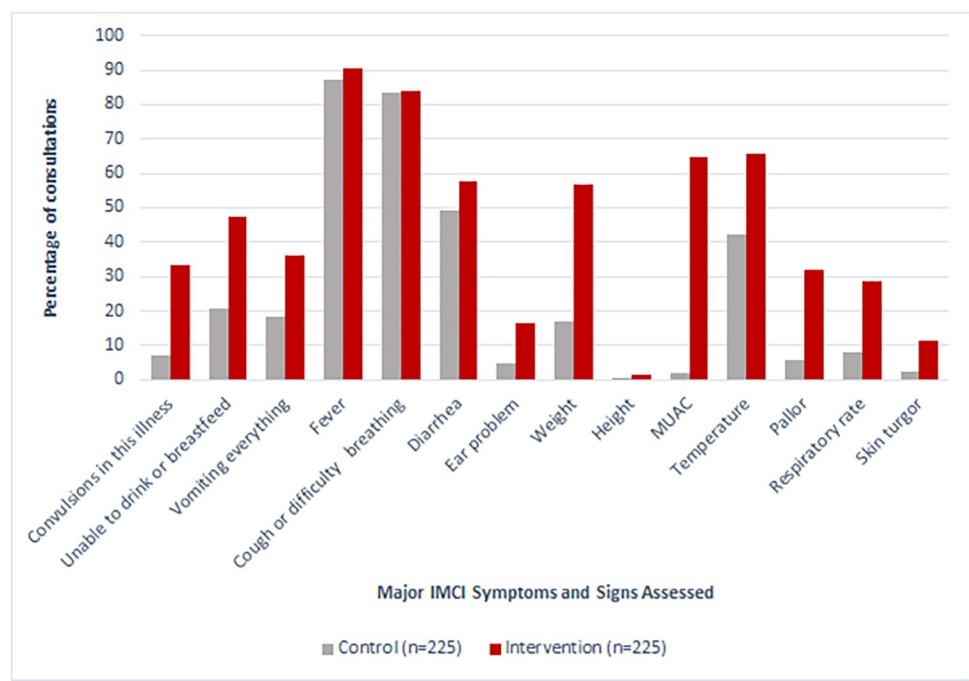

**Fig 4. Proportion of individual major IMCI symptoms and signs assessed.**

Table 3. Other Symptoms and Signs Assessed.

| Symptoms & signs assessed | Control, n/N (%) | Intervention, n/N (%) | Intraclass correlation coefficient | Adjusted risk ratio with 95% CI[a] | p-value |
|---|---|---|---|---|---|
| Lethargic or Unconscious[b] | 8/225 (3.6%) | 14/225 (6.2%) | 0.086 | 1.4 (0.6, 3.3) | 0.476 |
| Duration of fever[c] | 96/147 (65.3%) | 141/166 (84.9%) | 0.176 | 1.6 (0.9, 2.6) | 0.086 |
| Duration of cough[c] | 87/145 (60.0%) | 110/128 (85.9%) | 0.103 | 1.7 (1.1, 2.6) | **0.021** |
| Duration of diarrhea[c] | 28/39 (71.8%) | 43/46 (93.5%) | 0.107 | 1.3 (1.0, 1.6) | **0.036** |
| Mother's HIV status | 5/225 (2.2%) | 162/225 (72.0%) | 0.701 | 11.5 (4.1, 32.5) | **0.002** |
| Tuberculosis household contact | 2/225 (0.9%) | 60/225 (26.7%) | 0.422 | 1.9 (0.5, 7.7) | 0.361 |
| Neck stiffness | 2/147 (1.4%) | 2/166 (1.2%) | 0.010 | n/a[d] | |
| Felt behind ear[c] | 1/4 (25.0%) | 2/3 (66.7%) | n/a | n/a[d] | |
| Looked in mouth | 4/225 (1.8%) | 20/225 (8.9%) | 0.305 | 1.9 (0.5, 7.7) | 0.363 |
| Pulse oximetry | n/a | 24/164 (14.6%) | 0.717 | n/a | |
| Lung auscultation | 26/183 (14.2%) | 15/164 (9.2%) | 0.235 | 1.6 (0.5, 5.1) | 0.373 |
| Undressed the child | 43/225 (19.1%) | 99/225 (44.0%) | 0.379 | 3.4 (1.4; 8.3) | **0.011** |
| Checked health card | 68/225 (30.2%) | 89/225 (39.6%) | 0.552 | 1.5 (0.5, 5.0) | 0.465 |

CI: Confidence interval; HIV: Human Immunodeficiency Virus; n/a: not applicable

[a]The risk ratio was adjusted by type of health facility, council, healthcare worker cadre, healthcare worker years of experience, patient age and sex

[b]IMCI Danger sign

[c]Denominator based on patients for which clinically relevant to assess, i.e. duration of fever in those with reported fever; duration of cough/difficulty breathing, pulse oximetry or lung auscultation in those with cough or difficult breathing; duration of diarrhea in those with diarrhea; felt behind ear in those with an ear problem; neck stiffness in those with fever

[d]Adjusted risk ratio not calculated when fewer than 5 events

facilities) during the same 5-week period collected by the external clinical researchers in the same health facilities of the present analysis, was slightly lower than that observed by the external clinical researchers. The adjusted risk ratio remained nonetheless similar, 0.4, 95% CI 0.2 to 0.7, p-value 0.005.

81.9% of antibiotic prescriptions were appropriate in the intervention arm, compared to 51.4% in the control arm, adjusted relative risk of 1.5, 95% CI 1.2–1.5, p-value 0.003. All patients with malaria appropriately received an antimalarial treatment, and no patient without malaria received an antimalarial in both study arms.

Table 4. Counseling.

| Counseling topic | Control, n/N (%) | Intervention, n/N (%) | Intraclass correlation coefficient | Adjusted risk ratio with 95% CI[a] | p-value |
|---|---|---|---|---|---|
| Informed diagnosis | 92/225 (40.9%) | 105/225 (46.7%) | 0.454 | 1.4 (0.5, 4.0) | 0.557 |
| Feeding habit when not ill | 40/225 (17.8%) | 33/225 (14.7%) | 0.427 | 0.7 (0.2, 2.7) | 0.609 |
| Feeding when not ill | 32/225 (14.2%) | 27/225 (12.0%) | 0.258 | 0.7 (0.2, 1.8) | 0.391 |
| Extra fluids during current illness | 12/225 (5.3%) | 21/225 (9.3%) | 0.198 | 1.2 (0.4, 4.0) | 0.748 |
| Continue feeding and breastfeeding when ill | 25/225 (11.1%) | 33/225 (14.7%) | 0.292 | 1.0 (0.3, 3.1) | 0.987 |
| Danger signs to return to health facility | 13/225 (5.8%) | 17/225 (7.6%) | 0.152 | 0.6 (0.2, 1.3) | 0.182 |
| Discussed growth chart | 14/225 (6.2%) | 15/225 (6.7%) | 0.206 | 1.1 (0.3, 4.0) | 0.826 |
| Discuss follow up visit | 18/225 (8.0%) | 8/225 (3.6%) | 0.052 | 0.9 (0.4, 2.1) | 0.703 |
| Opportunity to ask questions | 87/225 (38.7%) | 122/225 (54.2%) | 0.717 | 1.0 (0.2, 4.9) | 0.998 |

CI: Confidence interval

[a]The relative risk was adjusted by type of health facility, council, healthcare worker cadre, healthcare worker years of experience, patient age and sex

**Table 5. Antibiotic prescription and appropriate antibiotic and antimalarial prescription.**

| | Control | Intervention | Intraclass correlation coefficient | Adjusted risk ratio with 95% CI | p-value |
|---|---|---|---|---|---|
| Antibiotic prescription as observed by external clinical researchers | 172/225 (76.4%) | 84/225 (37.3%) | 0.146 | 0.5 (0.4, 0.7) | **<0.001** |
| Antibiotic prescription as reported in ePOCT+ or eCRF by healthcare providers[ab] | 162/239 (67.8%) | 90/294 (30.6%) | 0.214 | 0.4 (0.2, 0.7) | **0.005** |
| Appropriate antibiotic prescription | 112/218 (51.4%) | 158/193 (81.9%) | 0.165 | 1.5 (1.2; 1.8) | **0.003** |
| Appropriate antimalarial prescription | 2/2 (100%) | 30/30 (100%) | n/a | n/a | **n/a** |

[a]Antibiotic prescription among new cases in patients aged 2–59 months, where antibiotic prescription was documented during the same days and same health facilities as the ancillary study

[b]Analysis performed using the same model including the same individual and cluster level adjustment factors, except for those related to the healthcare providers (education, and cadre)

## Discussion

This cluster randomized controlled trial in Tanzania found that the use of ePOCT+, a digital clinical decision support algorithm, for the management of sick children aged 2–59 months old, moderately increased (by 15%) the mean proportion of major IMCI symptoms and signs assessed by healthcare providers in primary care level health facilities. The overall proportion of IMCI symptoms and signs assessed at each individual consultation however remained low.

The overall increase in the mean proportion of major IMCI symptom and signs assessed by healthcare providers aligns with previous studies in Tanzania [13], Afghanistan [37], Nigeria [17], and Burkina Faso [16], but in contrast to findings in South Africa [42], which did not find improvements in the assessment of IMCI symptoms and signs. While these findings suggest that digital clinical decision support algorithms can positively enhance quality of care for

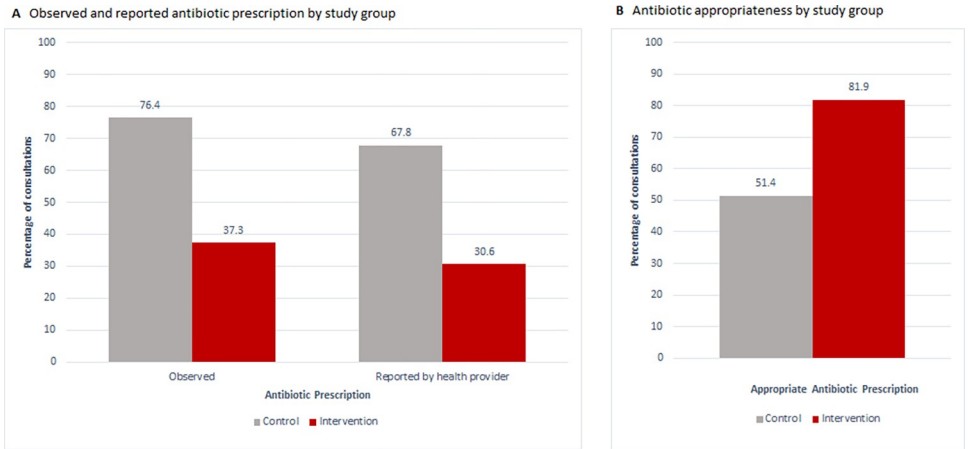

**Fig 5. Reported and observed antibiotic prescription and appropriateness.** Panel A presents the antibiotic prescription as observed by the external clinical researchers in 225 consultations in the control arm and 225 consultations in the intervention arm. It also presents the reported antibiotic prescription as documented by the healthcare providers in the ePOCT+ digital health tool among 239 new consultations in patients aged 2–59 months in intervention health facilities, and the documented antibiotic prescription within the eCRF in the control health facilities during the same days and same health facilities when the external clinical researchers were present. Panel B presents the appropriateness of antibiotic prescription according to the diagnoses documented or stated by the healthcare providers in the 225 control arm consultations and 225 intervention arm consultations.

sick children, there is much room for improvement. Indeed, the proportion of most individual IMCI symptoms and signs assessed in both control and intervention arms was low, much lower than in previous studies, and many symptoms and signs were not assessed more frequently in intervention health facilities. Differences compared to other studies, may be explained by the more pragmatic nature of the study compared to other studies conducted in more controlled settings (shorter pilot studies) where the Hawthorne effect may have a greater influence on healthcare provider behavior [13]. In addition, our study found much poorer adherence to IMCI in the control arm compared to most other similar studies; which could partly be explained by differences in the setting and healthcare providers, notably in the frequency and type of IMCI training provided to healthcare providers [16,17]. The low proportion of children assessed for danger signs in order to identify children at highest risk of mortality is most concerning (33% for convulsions, 48% for unable to drink or breastfeed, and 36% for vomiting everything in the intervention arm). Respiratory rate was also infrequently assessed (29% in intervention arm), despite being an essential sign to distinguish children with cough or difficulty breathing requiring antibiotics or not [30].

The statistically significant increase in the assessment of weight (aRR 4.9 [95% CI 1.9 to 12.9]), and MUAC (aRR 5.5 [95% CI 1.7 to 17.6]) is noteworthy, as they are critical anthropometric measures to identify children with malnutrition, a condition that contributes significantly to childhood morbidity and mortality. Systematically measuring weight and MUAC to identify and manage severe malnutrition can indeed improve clinical outcomes as well as the long-term health status of children [43]. The low proportion of children assessed for height/length reflects the difficulty and constraints of this measurement in particular [44], and the impact of not requiring the measurement to be mandatory within the digital tool.

While improved adherence to the assessment of IMCI symptoms and signs would likely be beneficial, translation to improved clinical outcomes should not automatically be assumed. Healthcare providers often integrate a number of clinical cues that may allow them to distinguish which child would truly require danger signs to be assessed, or respiratory rate to be measured. For example, a 2 year old child presenting to a primary care health facility smiling and playing in the consultation room, with complaints of cough and runny nose for the past 2 days without difficulty breathing or fever, is unlikely to have danger signs and very often not have fast breathing or chest indrawing. This raises the important question on how to best assess quality of care, and whether it can be done without assessment of clinical outcome.

The large variation in mean proportion of major IMCI symptoms and signs between intervention health facilities provide clues to higher potential benefits of the intervention. The three best health facilities have a mean score of 60% or above, and the three worst below 30%. While clinical decision support algorithms may help improve knowledge and information on what symptoms and signs to assess, it does not completely address the other barriers and bad habits linked to poor adherence to IMCI [9,10]. As with many complex health interventions, implementation of new interventions, and or guidelines may often not succeed with training alone, instead concomitant and meaningful mentorship must accompany it [45]. Indeed training, mentorship, and dashboards integrating benchmarking were part of the current intervention package, however these supportive interventions were not targeted towards assuring adherence to the assessment of many IMCI symptoms and signs but rather targeted on antibiotic stewardship, and overall uptake on the use of the tools. Adaptations to these supportive tools to target specific IMCI quality of care measures may help [46–48]. Further qualitative investigations are underway to better understand healthcare provider perspectives on barriers in adhering to ePOCT+ and the IMCI chartbook.

The study also revealed a two-fold reduction in antibiotic prescriptions (adjusted relative risk 0.5, 95% CI 0.4 to 0.7), and 50% improvement in the appropriate use of antibiotics in

health facilities using ePOCT+ (adjusted relative risk 1.5 (95% CI 1.2 to 1.8). These are critical findings given the global concern for bacterial antimicrobial resistance [49], and validating the results found in the parent trial [24]. Similar reductions were found in the parent trial using the intention-to-treat results over the full 11 month trial period (adjusted relative risk 0.6 (95% CI 0.5 to 0.6) [24], and in the same 5-week period as the present trial (adjusted relative risk 0.4, 95% CI 0.2 to 0.7). However, the documented antibiotic prescription as observed by external clinical researchers was slightly higher than that documented by the healthcare providers in ePOCT+ (intervention health facilities) and the eCRF (control health facilities), suggesting that some healthcare providers may under report antibiotic prescription in ePOCT+ and the eCRF.

There were several limitations to this study. First, the Hawthorne effect, the presence of an external clinical researcher observing the consultation may have influenced the healthcare provider's practice in both study arms. Indeed the uptake of the ePOCT+ tool in this study was higher compared to the parent trial (95% versus 76%), likely due to the presence of the researchers. Despite this, adherence to the IMCI chartbook was relatively low, and substantially lower compared to other studies, suggesting that the Hawthorne effect may not have had such a big impact, and the desirability bias minimal. Second, sample size; while the study was sufficiently powered for the primary outcome, interpretation of the secondary outcomes would have benefited from a higher sample size. Indeed many secondary outcomes did not show statistical significance despite relatively high effect size, likely due to the higher than expected heterogeneity between health facilities as indicated by the high ICC. Third, the intervention package included not only the digital tool, but also mentorship and benchmarking quality of care dashboards. It is thus not possible to understand what part, and to what extent the intervention package impacted quality of care. Finally as discussed in previous paragraphs, adherence to IMCI is an imperfect proxy for the measurement of quality of care.

In conclusion, a digital clinical decision support algorithm package can help improve quality of care, however adherence to IMCI remained low for many symptoms and signs in a close to real world assessment. Further efforts including innovative approaches to improve quality of care are highly needed. The implementation of multiple interventions, such as the development and improvement of supportive mentorship of clinicians, better healthcare provider incentives, task-shifting, ongoing training and performance accountability may help address the many barriers to quality of care.

## Supporting information

**S1 Checklist. CONSORT Checklist.**
(DOCX)

**S1 File. DYNAMIC Tanzania study protocol.**
(PDF)

**S2 File. Statistical analysis plan.**
(PDF)

## Acknowledgments

We would like to first thank all the participating healthcare providers, patients and caregivers. We acknowledge the contributions of the research assistants at the Ifakara Health Institute and Mbeya Medical Research Centre–National Institute of Medical Research, who assisted in the data collection, the Information Technology staff at Unisanté (Sylvain Schaufelberger, Greg Martin), and staff at Wavemind (Emmanuel Barchichat, Alain Fresco, and Quentin Girard)

for their work on the medAL-suite during the study, and Community Health Management Team members in the 5 participating councils in Tanzania for their collaboration in implementing the study. We acknowledge researchers of the Tools of Integrated Management of Childhood Illness and DYNAMIC Rwanda project for their contributions to ePOCT+ and the many common research activities (Dr Fenella Beynon, Dr Lena Matata, Dr Helene Langet, Dr Ludovico Cobuccio, Dr Victor Rwandacu, Dr Robert Moshiro). We would also like to acknowledge Janet Urquhart-Ducharme for her assistance in proofreading a version of this manuscript. We would like to acknowledge Dr Irene Masanja for her work on the initial development of the study, who regrettably passed away before the start of the study. Dr Godfrey Kavishe passed away before the submission of the final version of this manuscript. Rainer Tan accepts responsibility for the integrity and validity of the data collected and analyzed.

## Author Contributions

**Conceptualization:** Rainer Tan, Godfrey Kavishe, Alexandra V. Kulinkina, Lameck B. Luwanda, Kristina Keitel, Marie-Annick Le Pogam, Honorati Masanja, Valérie D'Acremont.

**Data curation:** Rainer Tan, Godfrey Kavishe, Sabine Renggli.

**Formal analysis:** Rainer Tan, Godfrey Kavishe.

**Funding acquisition:** Nyanda Ntinginya, Honorati Masanja, Valérie D'Acremont.

**Methodology:** Rainer Tan, Godfrey Kavishe, Marie-Annick Le Pogam.

**Project administration:** Godfrey Kavishe, Alexandra V. Kulinkina, Sabine Renggli, Lameck B. Luwanda, Chacha Mangu, Geofrey Ashery, Nyanda Ntinginya, Honorati Masanja.

**Software:** Rainer Tan, Godfrey Kavishe, Sabine Renggli, Ibrahim Evans Mtebene, Peter Agrea.

**Supervision:** Rainer Tan, Godfrey Kavishe, Alexandra V. Kulinkina, Sabine Renggli, Lameck B. Luwanda, Chacha Mangu, Geofrey Ashery, Margaret Jorram, Humphrey Mhagama, Marie-Annick Le Pogam, Nyanda Ntinginya, Honorati Masanja, Valérie D'Acremont.

**Writing – original draft:** Rainer Tan, Godfrey Kavishe.

**Writing – review & editing:** Rainer Tan, Godfrey Kavishe, Alexandra V. Kulinkina, Sabine Renggli, Lameck B. Luwanda, Chacha Mangu, Geofrey Ashery, Margaret Jorram, Ibrahim Evans Mtebene, Peter Agrea, Humphrey Mhagama, Kristina Keitel, Marie-Annick Le Pogam, Nyanda Ntinginya, Honorati Masanja, Valérie D'Acremont.

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
