## [Decision Letter · Decision Letter 0]

11 Sep 2024

PDIG-D-24-00255

A cluster randomized trial assessing the effect of a digital health algorithm on quality of care in Tanzania (DYNAMIC study)

PLOS Digital Health

Dear Dr. Tan,

Thank you for submitting your manuscript to PLOS Digital Health. After careful consideration, we feel that it has merit but does not fully meet PLOS Digital Health's publication criteria as it currently stands. Therefore, we invite you to submit a revised version of the manuscript that addresses the points raised during the review process.

Please submit your revised manuscript within 30 days Oct 11 2024 11:59PM. If you will need more time than this to complete your revisions, please reply to this message or contact the journal office at digitalhealth@plos.org. Please include the following items when submitting your revised manuscript:

We look forward to receiving your revised manuscript.

Kind regards,

Shannon Freeman, PhD

Academic Editor

PLOS Digital Health

Shannon Freeman

Academic Editor

PLOS Digital Health

Journal Requirements:

Additional Editor Comments (if provided):

This is a very interesting paper which will be of great interest to a wide readership both within Tanzania and internationally. The reviewers have provided some helpful suggestions for minor edits. Additionally, this paper would be strengthened with some careful proofreading and attention to spelling, grammar, and punctuation.

Reviewers' comments:

Reviewer's Responses to Questions

**Comments to the Author**

1. Does this manuscript meet PLOS Digital Health’s publication criteria? Is the manuscript technically sound, and do the data support the conclusions? The manuscript must describe methodologically and ethically rigorous research with conclusions that are appropriately drawn based on the data presented.

Reviewer #1: Yes

Reviewer #2: Partly

Reviewer #3: Yes

2. Has the statistical analysis been performed appropriately and rigorously?

Reviewer #1: Yes

Reviewer #2: Yes

Reviewer #3: Yes

3. Have the authors made all data underlying the findings in their manuscript fully available (please refer to the Data Availability Statement at the start of the manuscript PDF file)?

Reviewer #1: Yes

Reviewer #2: Yes

Reviewer #3: Yes

4. Is the manuscript presented in an intelligible fashion and written in standard English?

Reviewer #1: Yes

Reviewer #2: No

Reviewer #3: Yes

5. Review Comments to the Author

Reviewer #1: A topic and clinically relevant article concerning a cluster randomized trial assessing the effect of a digital health algorithm on quality of care in Tanzania (DYNAMIC study) indicating the potential value of such technologies.

Reviewer #2: 1.The study has certain significance and application value for the health management of a certain population with this digital CDSA .

2.Please indicate the detailed source of the statistical software, as well as the scope of application and reference standards of the statistical methods used in this study.

3. Some language writing (spelling and grammatical) needs to be corrected in the manuscript. Some sentences are too long and difficult to understand.

Reviewer #3: A cluster randomized trial assessing the effect of a digital health algorithm on quality of care in Tanzania (DYNAMIC study)," several areas need attention.

 First, the introduction should clearly articulate the specific research gap this study addresses and succinctly state the primary objectives and research questions. This will help readers understand the importance and relevance of the study within the broader field of digital health interventions. Additionally, more comprehensive details should be provided on the ePOCT+ digital clinical decision support algorithm (CDSA), including its functionalities and how it was integrated into routine care. This should also cover the training and mentorship components of the intervention, which are critical to understanding the intervention's implementation and effectiveness.

The statistical methods section would benefit from a more detailed explanation, including how missing data were handled and the rationale for using specific statistical models and covariates. This will enhance the robustness of the methodology and make it easier for readers to assess the validity of the findings. Furthermore, the presentation of results could be improved by using more visual aids, such as tables and figures, to summarize key outcomes and differences between the intervention and control groups. This approach would make the results more accessible and engaging.

In the discussion section, a deeper exploration of the study's findings and comparisons with similar studies would be valuable. The authors should provide potential explanations for the observed outcomes and discuss the implications for future digital health implementations. Moreover, a thorough discussion of the study's limitations, including potential biases and their impact on the findings, is necessary for transparency and to guide future research. Consistency in terminology and formatting throughout the manuscript will also enhance clarity and readability. Finally, a detailed data availability statement should be included to specify where and how the study data can be accessed, promoting transparency and reproducibility. Addressing these areas will significantly strengthen the manuscript, making it a more compelling and impactful contribution to the field of digital health.

6. PLOS authors have the option to publish the peer review history of their article (what does this mean?). If published, this will include your full peer review and any attached files.

**Do you want your identity to be public for this peer review?** For information about this choice, including consent withdrawal, please see our Privacy Policy.

Reviewer #1: No

Reviewer #2: Yes: Xiao-Yu Zhang

Reviewer #3: No

---

## [Decision Letter · Decision Letter 1]

7 Nov 2024

A cluster randomized trial assessing the effect of a digital health algorithm on quality of care in Tanzania (DYNAMIC study)

PDIG-D-24-00255R1

Dear Tan,

We are pleased to inform you that your manuscript 'A cluster randomized trial assessing the effect of a digital health algorithm on quality of care in Tanzania (DYNAMIC study)' has been provisionally accepted for publication in PLOS Digital Health.

Best regards,

Shannon Freeman, PhD

Academic Editor

PLOS Digital Health

**Additional Editor Comments (if provided):**

**Reviewer Comments (if any, and for reference):**

Reviewer's Responses to Questions

**Comments to the Author**

1. If the authors have adequately addressed your comments raised in a previous round of review and you feel that this manuscript is now acceptable for publication, you may indicate that here to bypass the “Comments to the Author” section, enter your conflict of interest statement in the “Confidential to Editor” section, and submit your "Accept" recommendation.

Reviewer #1: All comments have been addressed

Reviewer #2: All comments have been addressed

2. Does this manuscript meet PLOS Digital Health’s publication criteria? Is the manuscript technically sound, and do the data support the conclusions? The manuscript must describe methodologically and ethically rigorous research with conclusions that are appropriately drawn based on the data presented.

Reviewer #1: Yes

Reviewer #2: Yes

3. Has the statistical analysis been performed appropriately and rigorously?

Reviewer #1: Yes

Reviewer #2: Yes

4. Have the authors made all data underlying the findings in their manuscript fully available (please refer to the Data Availability Statement at the start of the manuscript PDF file)?

Reviewer #1: Yes

Reviewer #2: Yes

5. Is the manuscript presented in an intelligible fashion and written in standard English?

Reviewer #1: Yes

Reviewer #2: Yes

6. Review Comments to the Author

Reviewer #1: I am happy with the revised manuscript

Reviewer #2: (No Response)

7. PLOS authors have the option to publish the peer review history of their article (what does this mean?). If published, this will include your full peer review and any attached files.

**Do you want your identity to be public for this peer review?** For information about this choice, including consent withdrawal, please see our Privacy Policy.

Reviewer #1: No

Reviewer #2: None
